# Diagnostic Performance of Serum Erythropoietin to Discriminate Polycythemia Vera from Secondary Erythrocytosis through Established Subnormal Limits

**DOI:** 10.3390/diagnostics14171902

**Published:** 2024-08-29

**Authors:** Ji Sang Yoon, Hyunhye Kang, Dong Wook Jekarl, Sung-Eun Lee, Eun-Jee Oh

**Affiliations:** 1Department of Laboratory Medicine, Seoul St. Mary’s Hospital, College of Medicine, The Catholic University of Korea, Seoul 06591, Republic of Korea; gary431@naver.com (J.S.Y.); azuresky@hanmail.net (H.K.); bonokarl@catholic.ac.kr (D.W.J.); 2Research and Development Institute for In Vitro Diagnostic Medical Devices, College of Medicine, The Catholic University of Korea, Seoul 06591, Republic of Korea; 3Department of Hematology, Seoul St. Mary’s Hospital, College of Medicine, The Catholic University of Korea, Seoul 06591, Republic of Korea; lee86@catholic.ac.kr

**Keywords:** polycythemia vera, secondary erythrocytosis, serum erythropoietin, subnormal limit, reference interval, clinical decision limit, functional reference limit

## Abstract

Serum erythropoietin (sEPO) is an initial screening tool for distinguishing polycythemia vera (PV) from secondary erythrocytosis (SE), but defining ‘subnormal’ sEPO levels for PV diagnosis remains contentious, complicating its clinical utility. This study compares the diagnostic performance of sEPO across established subnormal limits, including reference interval (RI), clinical decision limit (CDL), and functional reference limit. sEPO levels were analyzed in 393 healthy donors (HDs) and 90 patients (41 PV and 49 SE), who underwent bone marrow biopsy and genetic tests due to erythrocytosis. The RI (2.5–97.5 percentile from HDs) of sEPO was 5.3–26.3 IU/L. A CDL of 3.1 IU/L, determined by ROC analysis in erythrocytosis patients, had a sensitivity of 80.5% and specificity of 87.8% for diagnosing PV. A functional reference limit of 7.0 IU/L, estimated based on the relationship between sEPO and hemoglobin, hematocrit, and WBC, increased sensitivity to 97.6% but decreased specificity to 46.7%. Using 5.3 IU/L as a ‘subnormal’ limit identified all three JAK2-negative PV cases, increasing the sensitivity and negative predictive value to 97.6% and 97.0%, respectively. Combining the RI, CDL, and functional reference limit may improve PV diagnostic accuracy.

## 1. Introduction

Serum erythropoietin (sEPO) is an endogenous glycoprotein hormone that regulates erythropoiesis by stimulating the formation and maturation of erythroid progenitors in bone marrow [1]. This 30.4 kDa molecule is primarily synthesized by the fetal liver and adult kidney with its production influenced by oxygen sensing in renal EPO-producing cells [1,2]. The main role of sEPO is to maintain red blood cell mass and hemoglobin (Hb) at stable levels. An elevated sEPO concentration may result from conditions, such as iron deficiency, anemia, myelodysplasia, or certain malignancies, while a decreased sEPO concentration may indicate renal diseases, polycythemia vera (PV), or myeloproliferative neoplasms (MPNs) [3].

Erythrocytosis may occur as a result of either an absolute or relative increase in red blood cell mass (RCM). Absolute erythrocytosis, characterized by an excessive production of red blood cells, can arise from clonal primary erythrocytosis or elevated sEPO secretion. Conversely, relative erythrocytosis shows increased hematocrit levels, but its RCM remains normal due to a reduction in plasma volume [4]. The most common form of primary acquired erythrocytosis is PV, which typically arises from the genetic mutation of JAK2 gene. Secondary erythrocytosis (SE) is a group of heterogeneous disorders, primarily triggered by abnormal sEPO activation in response to inadequate tissue oxygenation, such as cardiopulmonary disease, smoking, hydronephrosis, etc. Hypoxia-independent SE may also result from certain diuretic medications, post-renal transplantation, or autonomous sEPO production in certain tumors [5,6,7,8]. Although erythrocytosis may be associated with significant cardiovascular morbidity and thrombotic risk, PV carries an inherently higher risk of thrombosis and requires different treatment approaches [9,10]. In 2016, the World Health Organization (WHO) revised the diagnostic criteria for PV, which includes three major criteria, namely elevated Hb [>16.5 g/dL(men), >16.0 g/dL (women)] or hematocrit (Hct) [>49% (men), >48% (women)] specific for sex, hypercellularity with panmyelosis of the bone marrow, the presence of the JAK2 V617F or JAK2 exon 12 mutation, and one minor criterion of subnormal sEPO [11]. PV diagnosis requires meeting all three major criteria or the first two major criteria along with the minor criterion. Bone marrow biopsy may not be necessary in cases of sustained absolute erythrocytosis with Hb levels above 18.5 g/dL in men or above 16.5 g/dL in women, or Hct levels above 55.5% in men or above 49.5% in women, if JAK2 or JAK2 exon 12 mutation and the minor criterion are present. The initial step in erythrocytosis work up involves excluding the possibility of PV. Two key tests used for this purpose are the JAK2 mutation and sEPO analysis. Approximately 98 to 99% of PV patients harbor the JAK2 mutation in exon 12 or exon 14, and over 85% exhibit subnormal sEPO [5,12,13]. While the JAK2 mutation is a strong predictive marker for PV, genetic tests are limited due to long turnaround times and high costs [14,15]. In contrast, sEPO measurement offers a simple and rapid diagnostic approach for PV, with low sEPO being a potential indicator of the disease [5,15,16,17]. In cases where PV is suspected based on clinical findings and subnormal sEPO levels but JAK2 mutations are not detected, a bone marrow biopsy is recommended to evaluate histological features consistent with an MPN [18]. Despite its utility, the predictive value of a low sEPO remains controversial, and the diagnostic utility of EPO levels may be compromised by comorbidity, such as chronic obstructive pulmonary disease, smoking, and obesity [10,14,15,16,17,19,20]. This may be attributed to the lack of clarity in defining ‘subnormal’ sEPO for diagnosing PV, unlike the well-defined thresholds for Hb and Hct. Various studies have proposed different reference intervals (RIs) of sEPO, with the lowest lower limit being 2.2 IU/L and highest upper limit being 34.0 IU/L, thereby complicating its diagnostic use [21,22,23,24,25,26].

The most common approach in establishing RI involves calculating an RI on the central 95% of laboratory results from a disease-free reference population. However, RIs established in healthy populations are insufficient for interpreting subnormal sEPO [27]. Moreover, sEPO production in healthy individuals depends on factors, such as renal function, metabolic syndrome, inflammation, and aging [25,28]. The limitations of RI are emphasized in conditions where a significant proportion of the general population may have subclinical or benign diseases (e.g., SE for sEPO) that can be difficult to identify and exclude. An alternative approach is to determine the “functional” reference limit or RI estimation using both general and clinical populations through an indirect method [29,30]. Functional reference limits present clinically relevant reference limits across healthy, subclinical, and pathological populations, and they can be mathematically modeled from the relationship between interrelated biomarkers [30,31]. In contrast, clinical decision limits (CDLs) are defined as either the thresholds of diagnostic markers for the presence of a specific disease or association of a specific disease with the risk of an adverse outcome [30,32].

The aim of the present study is to establish a ‘subnormal’ limit for sEPO using RI in a healthy Korean population, CDL in a clinical population, and functional reference limit derived from laboratory data of both healthy and clinical populations. Furthermore, we aimed to compare the diagnostic performance using different subnormal sEPO limits in distinguishing PV from SE.

## 2. Materials and Methods

### 2.1. Study Design and Population

To establish an RI for sEPO, morning serum samples were collected from 393 healthy donors (HDs) who visited Seoul St. Mary’s Hospital for kidney allograft donation from January 2019 to March 2023. All donors underwent routine laboratory tests, and those meeting the donor criteria were enrolled. Donors with anemia, lung disease, or decreased renal function were excluded.

To assess diagnostic performance of sEPO in diagnosing PV, we retrospectively analyzed electronic medical records of patients who underwent bone marrow examination due to erythrocytosis within the same timeframe. Erythrocytosis was defined according to WHO diagnostic criteria for PV, Hb > 16.5 g/dL or Hct > 49% for men, and Hb > 16.0 g/dL or Hct > 48% for women. Only patients who received genetic tests regarding JAK2 V617F or exon 12 mutations had sEPO analyzed and underwent bone marrow biopsy were included. In addition, patients with anemia, pulmonary disorders (e.g., asthma or chronic obstructive pulmonary disease), decreased estimated glomerular filtration rate (eGFR) below 60 mL/min/1.73 m^2^, and those who received prior treatment such as phlebectomy or hydroxyurea were excluded due to reported effects on sEPO [33,34,35,36,37]. A total of 90 patients were included and categorized into PV group (*n* = 41) and SE group (*n* = 49) according to WHO diagnostic criteria [11].

This study was approved by the Institutional Review Board at Seoul St. Mary’s Hospital (KC19TESI0043 and KC23SISI0303). Written informed consent was obtained from all HDs for using their serum samples for research purposes. For patients with erythrocytosis, written informed consent was waived due to the retrospective nature of this study using medical records. Factors influencing endogenous sEPO were evaluated by investigating clinical and laboratory parameters, including complete blood count, blood chemistry, lipid profile, iron content, comorbidity, and smoking habit. All data were collected retrospectively from electronic medical records.

### 2.2. Serum Erythropoietin Assay

sEPO was measured using the Immulite^®^ 2000 EPO (Siemens Healthcare Diagnostics Inc., Tarrytown, NY, USA), which utilizes chemiluminescent method for detection. The assay consists of a ligand labeled monoclonal anti-EPO capture antibody, an alkaline phosphatase-labeled polyclonal conjugate antibody, and solid-phase anti-ligand-coated polystyrene beads. The assay showed an intra-assay variability of 3.4–8.8%, an inter-assay variability of 5.4–11.6% and a lower detection limit of 1.0 IU/L. Linearity was claimed by the manufacturer from 1.0 IU/L to 750 IU/L. Patients who had sEPO below 1.0 IU/L were considered to have sEPO of 1.0 IU/L [23,24]. The manufacturer suggested expected values as median of 10.6 IU/L and a central 95% range of 4.3–29.0 IU/L.

### 2.3. Statistical Analysis

Quantitative data were presented as median and interquartile range (IQR). To assess significant differences in sEPO among different groups, Kruskal–Wallis test and Mann–Whitney test were conducted. The RI for sEPO was expressed as the median and central 95% range (2.5th to 97.5th percentiles) [27,32]. Additionally, the 5th and 10th percentiles were utilized to identify supplementary lower limits, given the importance of detecting decreased sEPO for clinical decision making. Spearman’s rank correlation was performed to evaluate variables significantly affecting sEPO in the HD population. To determine CDL, receiver operating characteristic (ROC) analysis was performed, and the area under the curve (AUC) along with 95% confidence interval (CI) was measured using data from aforementioned 90 patients with erythrocytosis, which comprises PV group and SE group. The optimal cutoff for CDL was determined using Youden’s index, which is calculated by subtracting 1 from the sum of sensitivity and specificity. The functional reference limit was calculated by correlating sEPO with other laboratory biomarkers in a correlation graph visually displayed in the restricted cubit spine curve model. The functional reference limit was defined as the point where relative biomarkers started to show abrupt change in the correlation graph. A *p*-value < 0.05 was considered statistically significant. All statistical analyses were performed using SPSS version 24 (IBM Corporation, New York, NY, USA) and MedCalc Statistical Software version 20.114 (MedCalc Software Ltd., Ostend, Belgium). Figures were created using Prism GraphPad version 10.0.1 for windows (GraphPad Software, Boston, MA, USA).

## 3. Results

### 3.1. Basic Demographics of the Study Population

Table 1 presents the baseline characteristics of the 393 HDs and 90 patients with erythrocytosis. In the HD group, the median age was 51 years (range, 39–58 years), with 172 (43.8%) donors being male. The median (IQR) for Hb was 14.0 g/dL (13.2–15.2 g/dL). Among the 90 erythrocytosis patients, 41 patients (56.1% were male) were diagnosed with PV, and 49 patients (77.6% were male) were diagnosed with SE. The PV and SE patients had median (IQR) Hb of 18.7 g/dL (17.9–19.4 g/dL) and 18.4 g/dL (17.6–18.7 g/dL), respectively. Within the PV group, 37 patients had the JAK2 V617F mutation, 1 patient had the JAK2 exon 12 mutation, and 3 patients did not exhibit any mutations on the JAK2 gene.

Comparing the HD group to the SE group, the HD group had significantly lower blood pressure, heart rate, white blood cell (WBC), absolute neutrophil count (ANC), Hb, Hct, glucose, creatinine, and triglyceride compared to the SE group. Among patients with erythrocytosis, the PV group had higher WBC, ANC, and Hct, while the SE group had a higher frequency of smoking history and higher body weight, body mass index (BMI), glucose, and total cholesterol (*p* < 0.05).

### 3.2. Determination of Reference Interval and Influencing Factors for Serum Erythropoietin in Healthy Donors

To assess the impact of sex or age on sEPO in the HD group, we analyzed sEPO in sex- and age-based subgroups (<30, 30–39, 40–49, 50–59, ≥60 years). The median (IQR) sEPO in the age-based subgroups was 10.7 IU/L (9.2–13.9 IU/L) (<30 years), 9.4 IU/L (7.6–11.2 IU/L) (30–39 years), 9.7 IU/L (7.1–12.4 IU/L) (40–49 years), 10.7 IU/L (8.0–13.9 IU/L) (50–59 years), and 10.0 IU/L (8.4–12.7 IU/L) (≥60 years). No significant differences were observed among the sex- and age-based subgroups (Figure 1). In the 49 SE patients, sEPO levels did not differ significantly across subtypes, such as hypertension, diabetes, hyperlipidemia, and smoking (*p* > 0.05). Consequently, a single RI for sEPO was determined to be 5.3–26.3 IU/L based on the 95% central interval. The 2.5th, 5th, and 10th percentiles of sEPO in the HD group were 5.3 IU/L, 5.5 IU/L, and 6.4 IU/L, respectively. Spearman’s rank correlation analysis in healthy donors showed weak negative correlations with Hb, Hct, WBC, ANC, triglyceride, and diastolic blood pressure (*p* < 0.05) (Appendix A).

### 3.3. Determination of Clinical Decision Limit and Functional Reference Limit of Serum Erythropoietin

Comparing sEPO between the three groups, the HD group had the highest sEPO, followed by the SE group and the PV group [median (IQR); 10.1 IU/L (7.9–13.1 IU/L), 7.0 IU/L (3.8–11.2 IU/L), and 1.7 IU/L (1.1–3.0 IU/L), respectively] (*p* < 0.05). No significant difference in sEPO was observed between men and women in all three groups (*p* > 0.05) (Figure 2). The ROC curve for sEPO in patients with erythrocytosis yielded an AUC of 0.904 (95% CI 0.844–0.965). By calculating the Youden’s index, the optimal cutoff value for sEPO turned out to be 3.1 IU/L, which became the CDL (Figure 3).

Hb, Hct, and WBC were selected as sEPO-related biomarkers, and correlation graphs for each biomarker were drawn (Figure 4). The three biomarkers showed significant changes when sEPO was 7.0 IU/L, and the change became subtle when sEPO reached 10.0 IU/L. Therefore, the functional reference limit for sEPO was estimated to be 7.0 IU/L. All aforementioned subnormal limits (3.1 IU/L of CDL by ROC analysis, 4.3 IU/L expected value recommend by manufacturer, and 5.3 IU/L in the RI by 2.5th percentile of HD population) were lower than the estimated functional reference limit. We performed an additional analysis to validate the functional reference limit of sEPO, correlating Hb, Hct, and WBC in the SE and HD groups, excluding the PV group. The functional reference limit of sEPO (7.0 IU/L) showed sharp changes in Hb and Hct values, with more modest changes in WBC levels in the SE and HD groups (Appendix A). Additionally, to assess whether the functional reference limit of sEPO (7.0 IU/L) provides further insights into the SE and HD groups, we compared clinical and laboratory parameters based on this threshold. Subnormal sEPO levels (<7.0 IU/L) were associated with higher serum creatinine, elevated blood pressure, lower high-density lipoprotein (HDL) levels, and increased blood counts (*p* < 0.05) (Appendix A).

### 3.4. Diagnostic Performance of Serum Erythropoietin in Differential Diagnosis of Polycythemia Vera and Secondary Erythrocytosis Using Different ‘Subnormal’ Limits

Diagnostic performances based on the established ‘subnormal’ cutoff levels for sEPO are shown in Table 2. Using a CDL of 3.1 IU/L, the sensitivity, specificity, positive predictive value (PPV), and negative predictive value (NPV) were 80.5% (95% CI 65.1–91.2%), 87.8% (95% CI 75.2–95.4%), 84.6% (95% CI 71.9–92.2%), and 84.3% (95% CI 74.1–91.0%), respectively. When the manufacturer-recommended cutoff of 4.3 IU/L was used, sensitivity and NPV increased to 92.7% (95% CI 80.1–98.5%) and 92.3% (95% CI 79.9–97.3%), respectively. The cutoff level of 5.3 IU/L, based on the 2.5th percentile of the HD group, increased sensitivity and NPV up to 97.6% (95% CI 87.1–99.9%) and 97.0% (95% CI 82.0–99.6%), respectively, but reduced specificity and PPV down to 65.3% (95% CI 50.4–78.3%) and 70.2% (95% CI 61.5–77.6%), respectively. The functional reference limit of 7.0 IU/L had sensitivity of 97.6% (95% CI 87.1–99.9%), which was the same as that obtained using 5.3 IU/L as a cutoff. Three PV patients without mutation in JAK2 had an sEPO of 4.1, 3.5, and 1.4 IU/L, and these patients can be diagnosed as having subnormal sEPO using either RI or the functional reference limit.

## 4. Discussion

Considering recent interest in employing an indirect approach to explore correlation with laboratory data for conducting RI, we established various ‘subnormal’ sEPO limits to clarify its definition and predictive value [29,38]. The demographics of the 41 PV patients in the present study, with a male dominance (56.1%) and a median age of 56 years, are similar to those reported in previous studies [14,16]. Regarding the JAK2 mutation, 37 PV patients (90.2%) carried the V617F mutation, and 1 patient (2.4%) had an exon 12 mutation, while 49 SE patients did not have the JAK2 mutation. These results are consistent with previous reports [14,16,39].

The RI representing the 2.5th and 97.5th percentiles of sEPO measured in 393 HDs was determined to be 5.3–26.3 IU/L. In previous studies, nonparametric RIs using the same Immulite^®^ EPO assay in 170 healthy adults with a normal hematocrit ranged from 4.3 to 29.0 IU/L (median = 10.6 IU/L), and in 129 adults not screened for anemia, it was 3.3–23.4 IU/L [23,24]. Discrepancies in these results could be attributed to assay imprecision (6.4–10.3%) or differences in the study populations. Despite the fact that sEPO is influenced by various confounding factors, our study showed no significant effect of other clinical and laboratory factors, such as age, sex, smoking, and BMI, on sEPO [25,40,41].

Among factors that are significantly related with sEPO, Hb, Hct, and WBC are utilized in calculating the functional reference limit as they are associated with PV [13]. According to Sezgin et al., we can define the functional reference limit in many ways: sEPO when biomarkers show a significant change (7.0 IU/L), sEPO when biomarkers reach a plateau point (10.0 IU/L), etc. This is because no uniform rationale currently exists in defining the functional reference limit [30,31]. Previous findings suggest that EPO values above the midpoint of the reference range support SE [42]. Considering this fact, we presume 7.0 IU/L is a more adequate functional reference limit that represents the onset of polycythemia.

When we conducted an additional analysis of the functional reference limit in SE and HD, excluding the PV group, significant changes were observed in Hb and Hct levels. These findings suggest that the 7.0 IU/L threshold for sEPO is sensitive to variations in Hb and Hct in the SE and HD population, potentially reflecting its role in erythropoiesis regulation. Additionally, when comparing clinical and laboratory parameters in the SE and HD groups, subnormal sEPO levels (<7.0 IU/L) were associated with several adverse outcomes, including higher serum creatinine, elevated blood pressure, lower HDL levels, and increased blood counts. These associations may indicate an underlying link between low sEPO levels and systemic conditions contributing to cardiovascular and renal dysfunction. Functional reference limits may better reflect the underlying physiology of patients with erythrocytosis and may provide a basis for deriving a threshold related to anemia or erythrocytosis. However, for sEPO, a well-defined statistical definition of inflection points is lacking, and the relevant data we utilized in this study were limited. Therefore, we validated the ‘subnormal’ limits generated by RI and CDL in a restricted cubic spline model, and we found that the levels of ‘subnormal’ limits established in our study were lower than the functional reference limit. Further research is necessary to identify and validate factors that may affect sEPO.

Through ROC curve analysis, the AUC of sEPO was determined to be 0.904, indicating its efficacy as a diagnostic marker of PV. The CDL to discriminate PV from SE was established at 3.1 IU/L, with a sensitivity of 80.5% and specificity of 87.8%. Previous studies have reported variable sensitivities of sEPO with different cutoff levels, showing 12% sensitivity with a cutoff of 2.0 IU/mL [16], 68% sensitivity with a cutoff of 3.7 IU/L [43], and 82% sensitivity with a cutoff of 4.3 IU/L [15]. The sensitivity of CDL is probably due to the exclusion of the effects of previous phlebotomies, comorbidities, or other conditions reported to affect sEPO. This is important to consider when using sEPO as a parameter in PV diagnostic algorithms. Sensitivity can be improved to 97.6% using the lower limit of the RI (5.3 IU/L) because eight patients in the PV group had sEPO higher than 3.1 IU/L while only one patient had sEPO higher than 5.3 IU/L. Compared to the lower limit of the RI, the functional reference limit (7.0 IU/L) yielded the same sensitivity with lower specificity, PPV, and NPV. This discrepancy may be due to the small sample size in this study. A previous study found that subnormal EPO levels using a cutoff of 3.7 IU/L had a sensitivity of about 68% and a specificity of 94% [43]. Although EPO levels are specific, patients with PV can still have normal EPO levels, which could lead to misdiagnosis. Adjusting the cutoff could improve the sensitivity for diagnosing PV or distinguishing high-risk cases of SE, reducing the likelihood of missing PV cases. Further studies are required to confirm the optimal sEPO cutoff that maximizes diagnostic accuracy.

Our study benefitted from the inclusion of well-defined healthy individuals and patients with erythrocytosis. However, it also has limitations. We employed relatively small data sizes and excluded various factors that are known to affect sEPO, Hb, and Hct when selecting patients with erythrocytosis. Therefore, the functional reference limit may not fully represent the general population. Additionally, our data were collected from a single hospital in Korea. Larger studies, the replication of our methodology, and utilization of other methods and analyzers would provide a more comprehensive understanding of sEPO.

## 5. Conclusions

While sEPO alone may not be sufficient as a screening tool for PV diagnosis, it still holds potential in the diagnostic algorithm, especially for patients who do not have a JAK2 mutation or facilities where the JAK2 mutation test is unavailable. Through a novel approach based on the functional association between sEPO and other biomarkers, we validated ‘subnormal’ sEPO limits with clinical relevance to PV. We suggest that a sEPO concentration of 7.0 IU/L indicates the onset of polycythemia, and a 5.3 IU/L cutoff is an appropriate ‘subnormal’ concentration with higher sensitivity for PV diagnosis. In the future, determining the ‘subnormal’ limit of sEPO for diagnosing PV should be guided by global consensus.

## Figures and Tables

**Figure 1 diagnostics-14-01902-f001:**
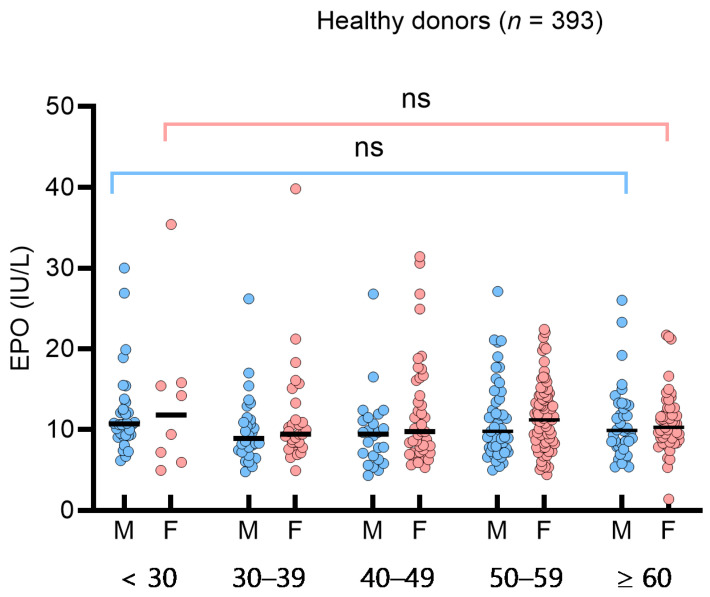
Serum erythropoietin (sEPO) in 393 healthy donors. sEPO was compared by male (blue circles) and female (red circles), and by age groups (<30 years, 30 to 39 years, 40 to 49 years, 50 to 59 years, and ≥60 years). Vertical lines represent interquartile ranges and the horizontal lines represent median. Abbreviations: ns, not significant.

**Figure 2 diagnostics-14-01902-f002:**
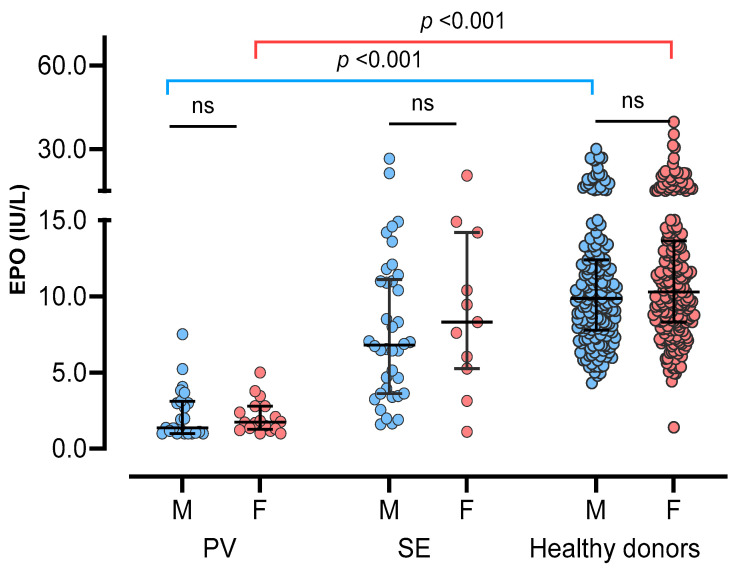
Serum erythropoietin in three study groups; polycythemia vera (PV), secondary erythrocytosis (SE), and healthy donors. Three groups were subdivided into male (blue circles) and female (red circles), and compared. Vertical lines represent interquartile ranges and the horizontal lines represent median. Abbreviations: ns, not significant.

**Figure 3 diagnostics-14-01902-f003:**
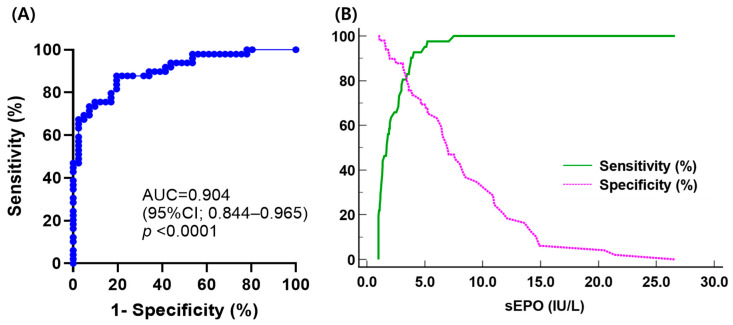
(**A**) Receiver operating characteristic curve analysis of serum erythropoietin to discriminate polycythemia vera (PV) from secondary erythrocytosis (SE); (**B**) plot of sensitivity and specificity versus criterion values in differentiating PV from SE.

**Figure 4 diagnostics-14-01902-f004:**
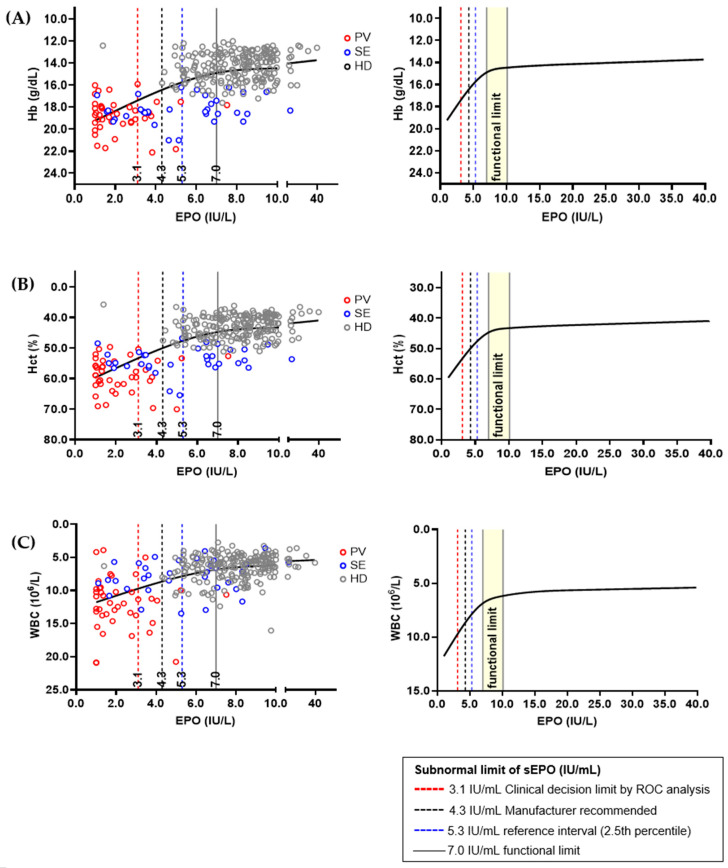
Relations between serum erythropoietin (sEPO) and interrelated biomarkers in different population; (**A**) hemoglobin (Hb), (**B**) hematocrit (Hct), and (**C**) white blood cell (WBC). The sEPO and three biomarkers from 393 healthy donors (HDs), 41 polycythemia vera (PV) patients and 49 secondary erythrocythemia (SE) patients were correlated using plot smoothing splines model. Functional reference limit of sEPO, which was 7.0 IU/L, is represented by a concentration showing dramatic change in Hb, Hct, and WBC value. Established ‘subnormal’ limits of sEPO are shown as dashed vertical lines, and their values were smaller than the estimated functional reference limit.

**Table 1 diagnostics-14-01902-t001:** Characteristics of the study population.

Characteristics	Healthy Donor (*n* = 393)	Polycythemia Vera (*n* = 41)	Secondary Erythrocytosis (*n* = 49)	*p*-Value (PV vs. SE)	*p*-Value (SE vs. HD)
Males, n (%)	172 (43.8)	23 (56.1)	38 (77.6)	0.031	<0.001
Age, years, median (range)	51 (19–77)	56 (26–89)	46 (18–74)	0.001	0.020
Body mass index ^1^, kg/m^2^	23.7 (22.0–26.4)	23.5 (21.4–26.4)	26.8 (24.4–28.9)	0.002	<0.001
Systolic BP ^1^, mmHg	122 (116–132)	131 (124–145)	136 (123–148)	0.443	<0.001
Diastolic BP ^1^, mmHg	76 (69–82)	89 (78–95)	89 (82–98)	0.192	<0.001
Heart rate ^1^, /min	74 (67–81)	81 (75–89)	86 (77–96)	0.229	<0.001
Smoking history, n (%)					
Yes	52 (13.2)	5 (12.2)	14 (28.6)	0.003	0.001
No	95 (24.2)	15 (36.6)	5 (10.2)	-	-
Unknown	246 (62.6)	21 (51.2)	30 (61.2)	-	-
Comorbidities, n (%)					
Hypertension	54 (13.7)	14 (34.1)	16 (32.7)	-	-
Diabetes Mellitus	9 (2.3)	4 (9.8)	6 (12.2)	-	-
Dyslipidemia	44 (11.2)	4 (9.8)	5 (10.2)	-	-
Pulmonary disease	0 (0.0)	0 (0.0)	0 (0.0)	-	-
Others	16 (4.1) ^2^	7 (17.1) ^3^	5 (10.2) ^4^	-	-
None	286 (72.8)	18 (43.9)	23 (46.9)	-	-
Laboratory data, median (IQR)					
Erythropoietin, IU/L	10.1 (7.9–13.1)	1.7 (1.1–3.0)	7.0 (3.8–11.2)	<0.001	<0.001
White blood cell, ×1000/μL	5.89 (4.82–7.03)	11.53 (9.88–13.77)	7.35 (5.82–9.23)	<0.001	<0.001
ANC, ×1000/μL	3.20 (2.44–4.10)	8.55 (6.73–10.79)	4.55 (3.12–6.19)	<0.001	<0.001
Hemoglobin, g/dL	14.0 (13.2–15.2)	18.7 (17.9–19.4)	18.4 (17.6–18.7)	0.102	<0.001
Hematocrit, %	42.1 (39.6–45.1)	58.9 (54.2–61.8)	53.7 (51.7–55.8)	<0.001	<0.001
Platelet, ×1000/μL	256 (226–292)	515 (352–700)	243 (204–288)	<0.001	0.201
Glucose, mg/dL	96 (91–103)	93 (87–103)	100 (90–130)	0.025	0.025
HbA1C, %	5.5 (5.3–5.7)	5.6 (5.4–6.2)	6.4 (5.6–7.7)	0.161	0.004
BUN, mg/dL	13.2 (11.0–15.5)	13.4 (10.9–16.0)	12.8 (10.1–15.7)	0.545	0.397
Creatinine, mg/dL	0.73 (0.64–0.85)	0.83 (0.72–0.94)	0.91 (0.76–1.04)	0.073	<0.001
eGFR, mL/min/1.73 m^2^	101 (95–110)	95 (81–103)	99 (84–111)	0.227	0.086
Total cholesterol, mg/dL	199 (172–223)	162 (141–182)	191 (160–218)	<0.001	0.304
Triglyceride, mg/dL	100 (69–149)	101 (79–150)	128 (82–168)	0.307	0.035
C-reactive protein, mg/dL	NT	0.07 (0.04–0.16)	0.07 (0.04–0.15)	0.972	-

^1^ Median (IQR), ^2^ angina, arrhythmia, cerebral vascular disease, fatty liver, gout, hyperthyroidism, hypothyroidism, and ulcerative colitis, ^3^ angina, arrhythmia, fatty liver, gout, iron deficiency, and myoma uteri, ^4^ angina, hyperthyroidism, hypopituitarism, and testosterone deficiency. Abbreviations: IQR, interquartile range; HD, healthy donor; PV, polycythemia vera; SE, secondary erythrocytosis; BP, blood pressure; ANC, absolute neutrophil count; BUN, blood urea nitrogen; eGFR, estimated glomerular filtration rate; NT, not tested.

**Table 2 diagnostics-14-01902-t002:** Diagnostic performance of serum erythropoietin levels with different cutoff levels.

Cutoff Value (IU/L)	Sensitivity, % (95% CI)	Specificity, % (95% CI)	PPV, % (95% CI)	NPV, % (95% CI)
3.1 *	80.5 (65.1–91.2)	87.8 (75.2–95.4)	84.6 (71.9–92.2)	84.3 (74.1–91.0)
4.3 ^†^	92.7 (80.1–98.5)	73.5 (58.9–85.1)	74.5 (64.5–82.4)	92.3 (79.9–97.3)
5.3 ^‡^	97.6 (87.1–99.9)	65.3 (50.4–78.3)	70.2 (61.5–77.6)	97.0 (82.0–99.6)
7.0 ^§^	97.6 (87.1–99.9)	46.9 (32.5–61.7)	60.6 (54.1–66.8)	95.8 (76.4–99.4)

* Cutoff level by ROC curve analysis in erythrocytosis patients; ^†^ cutoff level provided by the manufacturer; ^‡^ cutoff level of subnormal value determined by the reference interval in healthy donors (HD); ^§^ functional limit determined by correlating with related biomarkers in HD and erythrocytosis patients. Abbreviations: CI, confidence interval; PPV, positive predictive value; NPV, negative predictive value.

## Data Availability

The data presented in this study are available on request from the corresponding author.

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
