# Peer review of "Diagnostic Performance of Serum Erythropoietin to Discriminate Polycythemia Vera from Secondary Erythrocytosis through Established Subnormal Limits"

_diagnostics, 2024, doi:10.3390/diagnostics14171902_

Round 1
Reviewer 1 Report
Comments and Suggestions for Authors
The study by Sang Yoon Ji et al aimed to define the RI, CDL and the functional limits of sEPO in healthy donors, patients with secondary polycythemia and polycythemia vera from South Korea. The study is well conducted, nicely presented, and the English language is very good. My comments are as follows:
1. Line 39 should say„…renal disease or polycythemia vera, a myeloproliferative neoplams“ as eryhrocytosis in general may also be due to high sEPO
2. Line 40, erythrocytosis can also be „absolute“ (due to high red cell mass) or „relative“ (due to low plasma volume)
3. Line 46. I would emphasize the importance of differentiating PV from secondary polycythemia (SP) due to intrinsically higher thrombotic risk in PV and different treatment approaches. Also, for being completely up-to-date, I would also mention that SP may also bear thrombotic potential, especially in patients burdened with CV disease (see ie, doi: 10.1016/j.thromres.2021.11.025, doi: 10.1038/s41408-021-00463-x). The mentioned studies also complement this one with respect to similar patient characteristics in the PV and SP subgroups, supporting generalizability of the presented results.
4. Line 189 . No significant diference in sEPO between males and females but p <0.050?
5. Table 1. Are values in the brackets ranges or IQR? It seems its IQR.
6. Are there any differences in sEPO among different SP subtypes (i.e., smoking, pulmonary, kidney causes..?).
7. Do the authors have data about HD, SP, and PV patient follow-up? Do their survivals or thrombosis-free survivals differ? Perhaps the „functional“ sEPO limit may be used to differentiate HD and SP patients with inferior outcomes? This could support the authors conclusion about having „functional“ sEPO threshold. As it currently stands, it only shows that higher sEPO levels are associated with higher blood counts in all three subpopulations, which is plausible since sEPO is a growth hormone. However, I am unsure whether PV should also be included in this analysis since sEPO is supressed in PV because of clonal erythropoesis and its higher values may „drive“ the blood cell counts. In addition, Figure 2 shows significant overlaps in sEPO levels between SP and HD. In fact, some HD patients seem to have even higher sEPO values! Therefore line 262 that says that sEPO levels of 7.0 may represent better functional reference value for the onset of polycythemia does not seem supported by the data presented, as SP already have erythrocytosis - they have already been included as such in the study. I think the authors should 1) provide additional analysis for „functional“ sEPO levels with blood counts where PV patients are excluded from analysis, and 2) show that the „functional“ value of sEPO at 7.0 is, in addition to being associated with higher blood counts, associated with inferior outcomes in SP and HD. This would bring a totaly novel finding in SP patirents and HD where „functional“ sEPO levels may indeed serve as a novel prognostic biomarker. Currently, its role seems quite elusive.
Author Response
Reviewer #1:
The study by Sang Yoon Ji et al aimed to define the RI, CDL and the functional limits of sEPO in healthy donors, patients with secondary polycythemia and polycythemia vera from South Korea. The study is well conducted, nicely presented, and the English language is very good. My comments are as follows:
- Line 39 should say„…renal disease or polycythemia vera, a myeloproliferative neoplams“ as eryhrocytosis in general may also be due to high sEPO
Response; Thank you for careful review. Following the reviewer’s comment, we rephrased it as “while decreased sEPO concentration may indicate renal diseases or polycythemia vera (PV), a myeloproliferative neoplasms (MPN)” (lines 38-40).
- Line 40, erythrocytosis can also be „absolute“ (due to high red cell mass) or „relative“ (due to low plasma volume)
Response; Thank you for your valuable comments. Accordingly, we added sentence in the introduction as follows:
“Erythrocytosis may occur as a result of either an absolute or a relative increase in red blood cell mass (RCM). Absolute erythrocytosis, characterized by an excessive production of red blood cells, can arise from clonal primary erythrocytosis or elevated sEPO secretion. Conversely, relative erythrocytosis show increased hematocrit levels, but their RCM remains normal due to a reduction in plasma volume [4]” (lines 41 – 45)
- Line 46. I would emphasize the importance of differentiating PV from secondary polycythemia (SP) due to intrinsically higher thrombotic risk in PV and different treatment approaches. Also, for being completely up-to-date, I would also mention that SP may also bear thrombotic potential, especially in patients burdened with CV disease (see ie, doi: 10.1016/j.thromres.2021.11.025, doi: 10.1038/s41408-021-00463-x). The mentioned studies also complement this one with respect to similar patient characteristics in the PV and SP subgroups, supporting generalizability of the presented results.
Response; Thank you for the reviewer’s thoughtful feedback. In response to your suggestion, we have added the following passage and references to the introduction section.
“Although erythrocytosis may be associated with significant cardiovascular morbidity and thrombotic risk, PV carries an inherently higher risk of thrombosis and requires different treatment approaches [9,10].” (lines 51-54)
- Line 189 . No significant difference in sEPO between males and females but p <0.050?
Thank you for careful review. We have corrected it to ‘P>0.05’ (line 207)
- Table 1. Are values in the brackets ranges or IQR? It seems its IQR.
Thank you for your thoughtful review. We have clarified the values in Table 1 by including ‘median (range)’ or ‘median (IQR)’.
- Are there any differences in sEPO among different SP subtypes (i.e., smoking, pulmonary, kidney causes..?).
Response; Thank you for your valuable review. As per reviewer’ suggestion, we analyzed the sEPO levels across various subtypes of patients with secondary erythrocytosis, including hypertension, diabetes, hyperlipidemia, and smoking. However, our analysis revealed no significant differences in sEPO levels among these subtypes, likely due to the small sample size (n=49). We have updated the Results section accordingly with the following sentence:
“In the 49 SE patients, sEPO levels did not differ significantly across subtypes such as hypertension, diabetes, hyperlipidemia, and smoking (P > 0.05).” (lines 190-192)
- Do the authors have data about HD, SP, and PV patient follow-up? Do their survivals or thrombosis-free survivals differ? Perhaps the „functional“ sEPO limit may be used to differentiate HD and SP patients with inferior outcomes? This could support the authors conclusion about having „functional“ sEPO threshold. As it currently stands, it only shows that higher sEPO levels are associated with higher blood counts in all three subpopulations, which is plausible since sEPO is a growth hormone. However, I am unsure whether PV should also be included in this analysis since sEPO is supressed in PV because of clonal erythropoesis and its higher values may „drive“ the blood cell counts. In addition, Figure 2 shows significant overlaps in sEPO levels between SP and HD. In fact, some HD patients seem to have even higher sEPO values! Therefore line 262 that says that sEPO levels of 7.0 may represent better functional reference value for the onset of polycythemia does not seem supported by the data presented, as SP already have erythrocytosis - they have already been included as such in the study. I think the authors should 1) provide additional analysis for „functional“ sEPO levels with blood counts where PV patients are excluded from analysis, and 2) show that the „functional“ value of sEPO at 7.0 is, in addition to being associated with higher blood counts, associated with inferior outcomes in SP and HD. This would bring a totaly novel finding in SP patirents and HD where „functional“ sEPO levels may indeed serve as a novel prognostic biomarker. Currently, its role seems quite elusive.
Response; Thank you for your insightful comments. In response, we conducted an additional analysis to determine the functional reference limit of sEPO by correlating Hb, Hct, and WBC levels in the SE and HD groups, excluding the PV group (see Supplementary Figure 1). Furthermore, we compared clinical and laboratory parameters based on the functional limit of serum erythropoietin (7.0 IU/mL) in 49 patients with secondary erythrocytosis and 393 healthy donors (see Supplementary Table 2). We have updated the Results and Discussion section as follows:
“We performed an additional analysis to validate the functional reference limit of sEPO correlating Hb, Hct and WBC in the SE and HD groups, excluding the PV group. The functional reference limit of sEPO (7.0 IU/L) show sharp changes in Hb and Hct values, with more modest changes in WBC levels in SE and HD groups (supplementary Figure 1). Additionally, to assess whether the functional reference limit of sEPO (7.0 IU/L) provides further insights into the SE and HD groups, we compared clinical and laboratory parameters based on this threshold. Subnormal sEPO levels (< 7.0 IU/L) were associated with higher serum creatinine, elevated blood pressure, lower high-density lipoprotein (HDL) levels, and increased blood counts in HD and SE groups (P < 0.05) (Supplementary Table 2).” (in result section, lines 224-233)
“When we conducted an additional analysis of the functional reference limit in SE and HD, excluding the PV group, significant changes were observed in Hb and Hct levels. These findings suggest that the 7.0 IU/L threshold for sEPO is sensitive to variations in Hb and Hct in SE and HD population, potentially reflecting its role in erythropoiesis regulation. Additionally, when comparing clinical and laboratory parameters in the SE and HD groups, subnormal sEPO levels (< 7.0 IU/L) were associated with several adverse outcomes, including higher serum creatinine, elevated blood pressure, lower HDL levels, and increased blood counts. These associations may indicate an underlying link between low sEPO levels and systemic conditions contributing cardiovascular and renal dysfunction.” (in discussion section, lines 290-298)
Reviewer 2 Report
Comments and Suggestions for Authors
The article entitled “Diagnostic performance of serum erythropoietin to discriminate polycythemia vera from secondary erythrocytosis through established subnormal limits” is a retrospective study from a single hospital in Korea about the sensitivity and specificity of a subnormal EPO level in the diagnosis of PV od SE. There are few articles regarding this topic. Therefore, this information may be useful. EPO level is confounded by the presence of comorbidities such as obesity. I think that the authors should add this finding. In addition, the authors state that the cutoff of sEPO level of 5.3 IU/L or 7.9 IU/L have the same sensitivity. So the authors suggest a cutoff between 5.3-7.0 IU/L. This point need to be better detailed. Theefore, I think that this article is not suitable for publication in its current version.
Author Response
Reviewer #2:
The article entitled “Diagnostic performance of serum erythropoietin to discriminate polycythemia vera from secondary erythrocytosis through established subnormal limits” is a retrospective study from a single hospital in Korea about the sensitivity and specificity of a subnormal EPO level in the diagnosis of PV od SE. There are few articles regarding this topic. Therefore, this information may be useful. EPO level is confounded by the presence of comorbidities such as obesity. I think that the authors should add this finding. In addition, the authors state that the cutoff of sEPO level of 5.3 IU/L or 7.9 IU/L have the same sensitivity. So the authors suggest a cutoff between 5.3-7.0 IU/L. This point need to be better detailed. Therefore, I think that this article is not suitable for publication in its current version.
Response; Thank you for your insightful comments. In response, we have added the following passage and references to the introduction section.
“In cases where PV is suspected based on clinical findings, and subnormal sEPO levels but JAK2 mutations are not detected, a bone marrow biopsy is recommended to evaluate histological features consistent with a MPN [18]. Despite its utility, the predictive value of a low sEPO remains controversial and the diagnostic utility of EPO levels may be compromised by comorbidity such as chronic obstructive pulmonary disease, smoking, and obesity [14-17,19-21].” (lines 70-75)
In addition, according to the reviewer 1 and reviewer 2’s recommendations, we compared clinical and laboratory parameters based on the functional limit of serum erythropoietin (7.0 IU/mL) in SE and HD population (see Supplementary Table 2). We have also updated the Discussion section as follows.
“When we conducted an additional analysis of the functional reference limit in SE and HD, excluding the PV group, significant changes were observed in Hb and Hct levels. These findings suggest that the 7.0 IU/L threshold for sEPO is sensitive to variations in Hb and Hct in SE and HD population, potentially reflecting its role in erythropoiesis regulation. Additionally, when comparing clinical and laboratory parameters in the SE and HD groups, subnormal sEPO levels (< 7.0 IU/L) were associated with several adverse outcomes, including higher serum creatinine, elevated blood pressure, lower HDL levels, and increased blood counts. These associations may indicate an underlying link between low sEPO levels and systemic conditions contributing cardiovascular and renal dysfunction.” (in discussion section, lines 290-298)
" Compared to lower limit of the RI, the functional reference limit (7.0 IU/L) yielded the same sensitivity with lower specificity, PPV, and NPV. This discrepancy may be due to the small sample size of the study. A previous study found that subnormal EPO levels using a cutoff of 3.7 IU/L had a sensitivity of about 68% and a specificity of 94% [44]. Although EPO levels are specific, patients with PV can still have normal EPO levels, which could lead to misdiagnosis. Adjusting the cutoff could improve the sensitivity for diagnosing PV or distinguishing high-risk cases of SE, reducing the likelihood of missing PV cases. Further studies are required to confirm the optimal sEPO cutoff that maximizes diagnostic accuracy.” (in discussion, lines 317-325)
Reviewer 3 Report
Comments and Suggestions for Authors
This is an interesting manuscript addressing a clinically relevant issue. In fact, I recently had a referral of a patient described as polycythemia vera on the basis of a mildly elevated hemoglobin concentration, a JAK2 mutation allele frequency right at the lower edge of detectability, and a serum erythropoietin level that was higher than any of the lower limits described in this study but called low because it was not elevated.
There are some points where the authors can provide greater clarification:
1. In the description of the WHO diagnostic criteria, the authors need to note that under the 2016 criteria patients with more clearly elevated hemoglobin concentration (16.5 g/dL female/18.5 g/dL male), JAK2 mutation, and a low serum erythropoietin would be considered to be polycythemia vera.
2. In Section 2.1 , were people with erythrocytosis excluded from the healthy donor population? It does not say so specifically as it does for anemia.
3. In Table 1, the definition of "alcohol history" is not stated and is problematic as described by criteria of Exist/Never/Unknown which is the vast majority. The authors either need to define what they consider positive alcohol history – as written, one could presume that having taken a single drink in one's life constituted an existing alcohol history. Since it is unknown for the vast majority of subjects and is not statistically significant, it might be best to just delete.
4. Unlike alcohol history, smoking history is significant because of its association with secondary erythrocytosis. I would suggest that the authors change the terminology to yes/no as opposed to exist/never.
5. The authors note that white blood count has a week negative correlation with serum erythropoietin level, as do triglyceride and diastolic blood pressure levels. For white count at least, it may be worth noting in the manuscript that leukocytosis and particularly an increase in neutrophils, is a common feature of polycythemia vera.
6. The manuscript would benefit from a more detailed and explicit description of how the functional limit and clinical decision limit were determined. In particular, a description of what Youden's index represents functionally and how it is determined would be helpful.
Author Response
Reviewer #3:
This is an interesting manuscript addressing a clinically relevant issue. In fact, I recently had a referral of a patient described as polycythemia vera on the basis of a mildly elevated hemoglobin concentration, a JAK2 mutation allele frequency right at the lower edge of detectability, and a serum erythropoietin level that was higher than any of the lower limits described in this study but called low because it was not elevated.
There are some points where the authors can provide greater clarification:
- In the description of the WHO diagnostic criteria, the authors need to note that under the 2016 criteria patients with more clearly elevated hemoglobin concentration (16.5 g/dL female/18.5 g/dL male), JAK2 mutation, and a low serum erythropoietin would be considered to be polycythemia vera.
Response; Thank you for careful review. Taking your advice, we’ve revised the manuscript as follows.
“Bone marrow biopsy may not be necessary in cases of sustained absolute erythrocytosis with Hb level above 18.5 g/dL in men or above 16.5 g/dL in women, or Hct levels above 55.5% in men or above 49.5% in women, if JAK2 or JAK2 exon 12 mutation and the minor criterion are present.” (lines 60-63)
- In Section 2.1 , were people with erythrocytosis excluded from the healthy donor population? It does not say so specifically as it does for anemia.
Response; Thank you for your through review. As described in 2.1 study design and population section, erythrocytosis was not excluded when selecting the healthy donor population; “Donors with anemia, lung disease or decreased renal function were excluded.” (in lines 105-106). Upon reviewing the data, while some healthy donors had relatively high hemoglobin or hematocrit levels, none met the criteria for absolute erythrocytosis.
In Table 1, the definition of "alcohol history" is not stated and is problematic as described by criteria of Exist/Never/Unknown which is the vast majority. The authors either need to define what they consider positive alcohol history – as written, one could presume that having taken a single drink in one's life constituted an existing alcohol history. Since it is unknown for the vast majority of subjects and is not statistically significant, it might be best to just delete.
Response; We sincerely appreciate your comment and suggestion. In response to the mentioned point, we deleted the alcohol history data in Table 1.
- Unlike alcohol history, smoking history is significant because of its association with secondary erythrocytosis. I would suggest that the authors change the terminology to yes/no as opposed to exist/never.
Response; We sincerely appreciate your comment and suggestion. In response to the mentioned point, we modified the classification of smoking history as yes/no on Table 1.
- The authors note that white blood count has a week negative correlation with serum erythropoietin level, as do triglyceride and diastolic blood pressure levels. For white count at least, it may be worth noting in the manuscript that leukocytosis and particularly an increase in neutrophils, is a common feature of polycythemia vera.
Response; Thank you for your through review. The findings related to leukocytosis in the PV group are detailed in the Results section (3.1. Basic Demographics of the Study Population), where we note that “PV group had higher WBC, ANC, and Hct, while SE group had a higher frequency of smoking history and higher body weight, body mass index (BMI), glucose and total cholesterol (P < 0.05).” (in lines 174-176)
The description of the weak negative correlations between sEPO and variables such as Hb, Hct, WBC, ANC, triglyceride, and diastolic blood pressure in the Results 3.2 section specifically refer to the HD (healthy donor) population. Based on your suggestion, we have clarified this in the manuscript as follows:
“Spearman’s rank correlation analysis in healthy donors showed a negative correlation of sEPO with Hb, Hct, WBC, ANC, triglyceride and diastolic blood pressure (P<0.05)” (lines 194-196)
- The manuscript would benefit from a more detailed and explicit description of how the functional limit and clinical decision limit were determined. In particular, a description of what Youden's index represents functionally and how it is determined would be helpful.
Response; We sincerely thank the reviewer for thoughtful feedback. Considering the reviewer’s comments, we have added a more detailed explanation of Youden’s index and how the CDL and functional reference limit were determined as follows.
“The optimal cutoff for CDL was determined using Youden’s index, which is calculated by subtracting 1 from the sum of sensitivity and specificity. The functional reference limit was calculated by correlating sEPO with other laboratory biomarkers in a correlation graph visually displayed in the restricted cubit spine curve model. The functional reference limit was defined as the point where relative biomarkers started to show abrupt change in the correlation graph.” (lines 150-155)
Round 2
Reviewer 1 Report
Comments and Suggestions for Authors
Thanks for addressing all issues. I have no more comments.